# ON DYADIC FAIRNESS: EXPLORING AND MITIGATING BIAS IN GRAPH CONNECTIONS

**Peizhao Li[1], Yifei Wang[1], Han Zhao[2], Pengyu Hong[1], Hongfu Liu[1]**
[1]Brandeis University, [2]University of Illinois at Urbana-Champaign
`{peizhaoli,yifeiwang,hongpeng,hongfuliu}@brandeis.edu`
`hanzhao@illinois.edu`

## ABSTRACT

Disparate impact has raised serious concerns in machine learning applications and its societal impacts. In response to the need of mitigating discrimination, fairness has been regarded as a crucial property in algorithmic designs. In this work, we study the problem of disparate impact on graph-structured data. Specifically, we focus on dyadic fairness, which articulates a fairness concept that a predictive relationship between two instances should be independent of the sensitive attributes. Based on this, we theoretically relate the graph connections to dyadic fairness on link predictive scores in learning graph neural networks, and reveal that regulating weights on existing edges in a graph contributes to dyadic fairness conditionally. Subsequently, we propose our algorithm, **FairAdj**, to empirically learn a fair adjacency matrix with proper graph structural constraints for fair link prediction, and in the meanwhile preserve predictive accuracy as much as possible. Empirical validation demonstrates that our method delivers effective dyadic fairness in terms of various statistics, and at the same time enjoys a favorable fairness-utility tradeoff.

## 1 INTRODUCTION

The scale of graph-structured data has grown explosively across disciplines (e.g., social networks, telecommunication networks, and citation networks), calling for robust computational techniques to model, discover, and extract complex structural patterns hidden in big graph data. Research work has been proposed for inference learning on potential connections (Liben-Nowell & Kleinberg, 2007), and corresponding algorithms can be used for high-quality link prediction and recommendations (Adamic & Adar, 2003; Sarwar et al., 2001; Qi et al., 2006). In this work, we study the potential disparate impact in the prediction of dyadic relationships between two instances within a homogeneous graph.

Despite the wide applications of link prediction algorithms, serious concerns raised by disparate impact (Angwin et al., 2016; Barocas & Selbst, 2016; Bose & Hamilton, 2019a; Liao et al., 2020) should also be reckoned with by algorithm designers. In an algorithmic context, disparate impact often describes the disparity in influential decisions which essentially derives from the characteristics protected by anti-discrimination laws or social norms. Unfortunately, this negative impact derived from biased data and conventional algorithms occurs in many applications including link prediction. One example is that a user recommender system follows the proximity principle (individuals are more likely to interact with similar individuals) or existing connections with intrinsic bias. Such an operating mode would deliver biased recommendations dominated by sensitive attributes. For example, users with the same religion or ethnic group are more likely to be recommended to a user, and consequently generate segregation in social relations by long-term accumulation (Hofstra et al., 2017). Another example can be noticed in news streaming. When a news app has collected the political profile from a user, in pursuit of the user preference in news streaming, the system might only deliver politicking that the user is predisposed to agree with, therefore skews a user's scope and narrows the view by selectively displaying reality (Pariser, 2011). To alleviate these concerns, an algorithm should perform a link prediction without being biased by the sensitive attribute of the two instances, and should also stream diverse and preferred recommendations.

Motivated by the potential bias in real cases, in this paper we propose dyadic fairness for the link prediction problem in homogeneous graphs, where the dyadic fairness criterion expects the predictions

to be statistically independent of the sensitive attributes from the given two vertices. We focus our scope on Graph Neural Networks (GNNs), which have already shown remarkable capacity in graph representation learning by message passing along the graph structure (Xu et al., 2018; 2020; Ying et al., 2018; Wang et al., 2019; Fan et al., 2019; Li et al., 2020). Within the pipeline of GNNs, given an arbitrary graph, we theoretically analyze the relationship between dyadic fairness and the graph connections. Our findings suggest adapting weights on existing edges in a graph can contribute to dyadic fairness conditionally. Continuing with our theoretical findings, we propose **FairAdj**, an algorithm to empirically learn a fair adjacency matrix by updating the normalized adjacency matrix while keeping the original graph structure unchanged. Integrating with a utility objective function, the proposed algorithm seeks supplied dyadic fairness and link predictive utility simultaneously.

Our definition of dyadic fairness in a graph context is inspired by the statistical metrics in group fairness (Dwork et al., 2012; Kusner et al., 2017). First, vertices in a graph are categorized into several groups according to a protected attribute. Then, the dyadic fairness criterion asks some standard statistics such as positive outcomes or false positive rate on link score to be approximately equalized across intra and inter groups. Essentially, such a requirement asks for a more diverse prediction between and within different groups defined by the protected attribute, hence it also allows to mitigate social segregation by asking for more interactions across different protected groups in the graph.

Empirically, we present studies on six real-world social and citation networks to demonstrate the effectiveness of the proposed method. We conduct evaluations towards seven measurements of both utility and dyadic fairness. Comparing to other baseline methods (Kipf & Welling, 2016b; Grover & Leskovec, 2016; Rahman et al., 2019; Bose & Hamilton, 2019b), we consistently observe improvements from two aspects. First, dyadic fairness metrics verify that our method can minimize the statistical gap between the predictions of intra and inter links. Second, in terms of utility, our results are consistent with the existing literature (Zhao & Gordon, 2019; Fish et al., 2016; Calders et al., 2009), that satisfying fairness can potentially lead to a decrease in utility. However, our algorithm enjoys a more favorable fairness-utility tradeoff (same in fairness but less sacrifice in utility, and vice versa) when compared to previous works. Additionally, to approach the real application cases, we also showcase a direct product that comes from dyadic fairness: our method can effectively stream more diverse recommendations containing instances holding different kinds of sensitive attributes.

## 2 RELATED WORK

In this section we mainly review some closely related work in both fair machine learning and graph representation learning. We also briefly describe and discuss several existing works on learning fair node representations.

**Fair Machine Learning.**   Various types of fairness notions have been proposed and studied, including group fairness (Kusner et al., 2017; Kearns et al., 2018; 2019), individual fairness (Dwork et al., 2012), and preference-based notions (Zafar et al., 2017a; Ustun et al., 2019). Embracing these definitions, relevant algorithms involving fair constraints have been proposed. Zemel et al. (2013) propose a method to find a good representation to maximize utility while preserving both group and individual fairness. Following works on fair representation learning use autoencoder (Madras et al., 2018) or adversarial training (Zhao & Gordon, 2019; Zhao et al., 2019; Edwards & Storkey, 2015; Louizos et al., 2016) to simultaneously remove the sensitive patterns while preserving enough information for prediction. Zafar et al. (2017b) optimize for decision boundary fairness through regularization in logistic regression and support vector machines, and some other works achieve fairness by optimal transport between sensitive groups (Gordaliza et al., 2019; Jiang et al., 2019) and fair kernel methods (Donini et al., 2018). However, most proposed learning algorithms for fairness are mainly built on independent and identically distributed data, which are not suitable to be directly applied to graph-structured data with dyadic fairness.

**Graph Representation Learning.**   Representation learning on graphs is formulated to convert a structural graph into a low-dimensional space while preserving the discriminative and structural representations. Efficient graph analytic methods (Von Luxburg, 2007; Tang et al., 2015; Perozzi et al., 2014; Grover & Leskovec, 2016; Xu et al., 2019) can benefit a series of downstream applications including node classification (Wang et al., 2017), node clustering (Nie et al., 2017), link

prediction (Zhang & Chen, 2018) and graph classification as well. Recently, Graph Neural Networks (GNNs) have shown remarkable capacity in graph representation learning, with emergent varieties (Kipf & Welling, 2016a; Veličković et al., 2017; Hamilton et al., 2017) consistently delivering promising results. Our work uses GNNs for graph representation learning but targets improving dyadic fairness in link prediction.

**Fair Graph Embedding.** As fairness in graph-structured data a relatively new topic for research, only a few studies have investigated the fair issues in graph representation learning. Rahman et al. (2019) first proposed Fairwalk, a random walk based graph embedding method that revises the transition probability according to the vertex's sensitive attributes. Following the idea of adversarially removing sensitive patterns (Madras et al., 2018), Liao et al. (2020) proposed to use adversarial training on vertex representations to minimize the marginal discrepancy. This work mainly focuses on learning node representations that are free of sensitive attributes, which is different from ours. Other works includes fair collaborative filtering (Yao & Huang, 2017), item recommendation (Steck, 2018; Chakraborty et al., 2019) in bipartite graphs, and fair graph covering problem (Rahmattalabi et al., 2019).

## 3 PRELIMINARIES

Let $\mathcal{G} := (\mathcal{V}, \mathcal{E})$ as a graph with a fix set of vertices $\mathcal{V}$ and edges $\mathcal{E}$, where vertex features with $M$ dimensions are represented by $X \in \mathbb{R}^{N \times M}$. A nonnegative adjacency matrix $A \in \mathbb{R}^{N \times N}$ describes the relations between every pair of vertices. The element $a_{vu}$ in $A$ represents the weight on the linkage bridging $v$ and $u$, and is set to zero if no link exists. Every vertex holds a sensitive attribute, and we use $S(v)$ to denotes the sensitive attribute as well as the sensitive group membership of $v$. Let $\Gamma(v)$ be the set of 1-hop neighbors of $v$ including self-loop. Edge $(v, u)$ is called intra if $S(v) = S(u)$, and inter implies $S(v) \neq S(u)$. $|S|$ denotes the cardinality of group $S$. For a binary sensitive attribute with two groups $S_0$ and $S_1$ separated from the graph, $\widetilde{S_0} := \{v \in S_0 \mid \Gamma(v) \cap S_1 \neq \varnothing\}$ represents the set of vertices in $S_0$ which locate on the boundary and has connections with $S_1$, and the same for $\widetilde{S_1}$. Set $U$ to be the discrete uniform distribution over the set of vertices $\mathcal{V}$. Suppose a bivariate link prediction function $g(\cdot, \cdot) : \mathbb{R}^D \times \mathbb{R}^D \to \mathbb{R}$, that given two vectors of the embedded vertices representations, a value is obtained showing the model belief that these two vertices are potentially linked.

Having these basic notations, we consider the disparity in link prediction bridging on intra and inter sensitive groups. The general purpose of dyadic fairness is to predict links independently of whether two vertices having the same sensitive attribute or not. We extend from demographic parity (Edwards & Storkey, 2015; Kipf & Welling, 2016b; Madras et al., 2018; Zemel et al., 2013) to formulate a specific criteria for dyadic fairness. In a binary classification problem, demographic parity expects a classifier gives positive outcomes to two sensitive groups at the same rate. We turn the two groups in the content of demographic parity into the groups of intra and inter links. Ideally, achieving dyadic fairness will bring intra and inter link predictions at the same rate from a bag of candidate links.

Having vertices representation $v$ and $u$, dyadic fairness can be mathematically formulated as

**Definition 3.1.** A link prediction algorithm satisfies dyadic fairness if the predictive score satisfy

$$\Pr(g(u, v) | S(u) = S(v)) = \Pr(g(u, v) | S(u) \neq S(v)) \tag{1}$$

To quantify the fairness, we establish dyadic fairness on link prediction upon a fixed set of vertices, and models the expectation of absolute difference in score outcome across the groups of intra and inter links. Note that we also comprehensively evaluate our model on fairness by other four statistics gap extended from (Hardt et al., 2016) in Section 6.

## 4 HOW GRAPH CONNECTIONS AFFECT FAIRNESS

In this section, we propose a chain of theoretical analyses[1] established on a variant of demographic parity and graph neural networks to associate dyadic fairness with graph connections. We first

---

[1]Proofs for the proposition and theorem are in Appendix A.

demonstrate demographic parity in the outcomes of link prediction can be sufficiently reduced to the achievement of fair vertex representations when employing an inner product function for link prediction. Suggested by the sufficiency, we reveal how the pipeline of a one-layer graph neural network can affect the demographic parity, and draw the conclusion that for an arbitrary graph using GNNs for embedding, properly regulating the weights on existing graph connections can contribute to fairness conditionally. The theoretical findings motivate our algorithmic design as presented in the next section. Without loss of generality, in this section, we consider the sensitive attribute to be binary, where two sensitive groups $S_0$ and $S_1$ can be separated from the graph, but show the cases with sensitive attributes in multiple categorical values in our experimental section.

**Proposition 4.1.** For a link prediction function $g(\cdot, \cdot)$ modeled as inner product $g(v, u) = v^\top \Sigma u$, where $\Sigma \in \mathbb{S}_{++}^M$ is a positive-definite matrix, $\exists Q > 0, \forall v \sim \mathcal{V}, \|v\|_2 \leq Q$, for $\mathbb{E}_{v \sim U}[v] \in \mathbb{R}^M$, for dyadic fairness based on demographic parity, if $\|\mathbb{E}_{v \sim U}[v \mid v \in S_0] - \mathbb{E}_{v \sim U}[v \mid v \in S_1]\|_2 \leq \delta$,

$$\Delta_{\mathrm{DP}} := |\mathbb{E}_{(v,u) \sim U \times U}[g(v, u) \mid S(v) = S(u)] - \mathbb{E}_{(v,u) \sim U \times U}[g(v, u) \mid S(v) \neq S(u)]| \leq Q\|\Sigma\|_2 \cdot \delta. \tag{2}$$

**Remark 1.** Proposition 4.1 can be applied for a general inner product function in Euclidean space, where $\Sigma$ directionally and differently scales two input vectors. When setting $\Sigma$ to an identity matrix, function $g(\cdot, \cdot)$ reduces to dot product and is widely used in a series of research work on link prediction (Kipf & Welling, 2016b; Trouillon et al., 2016; Yao & Huang, 2017).

The above proposition implies fair vertex representations is a sufficient condition to achieve demographic parity in link prediction. Suggested by the sufficiency, the approach to fairness could be reduced to achieving vertex representations with a small discrepancy between sensitive groups. With the proposition, we are ready to proceed to understand fairness within graph neural networks and reveal how the structure or connections of a graph could affect demographic parity.

A single layer GNN can be generically written as $\mathrm{GNN}_\theta(X, \widetilde{A}) := \rho(\widetilde{A} X W_\theta)$, where $\rho$ is a non-linear activation function, $\widetilde{A}$ is the normalized adjacency matrix, and $W_\theta$ is the trainable weight matrix. One GNN layer can be decomposed into two disjoint phases: a vertex feature smoothing phase over the graph using $\widetilde{A}$, and a feature embedding phase using $W_\theta$ and $\rho$. Concretely, we consider left normalization $\widetilde{A} = D^{-1} A$ ($D$ is the degree matrix) for feature smoothing. Equivalently, at an individual level, for each vertex it is one-hop mean-aggregation $\mathrm{Agg}(v) := \deg_w(v)^{-1} \sum_{u \in \Gamma(v)} a_{vu} u$, where $\deg_w(v) := \sum_{u \in \Gamma(v)} a_{vu}$ stands for the weighted degree of vertex $v$.

We respectively abbreviate $\mathbb{E}_{v \sim U}[v | v \in S_0]$ and $\mathbb{E}_{v \sim U}[v | v \in S_1]$ as $\mu_0$ and $\mu_1$. Let $\sigma$ denotes the maximal deviation of vertex representations, namely, $\forall v \in S_0, \|v - \mu_0\|_\infty \leq \sigma$, and $\forall v \in S_1$, $\|v - \mu_1\|_\infty \leq \sigma$. Let $D_{\max} := \max_{v \in \mathcal{V}} \deg_w(v)$ be the maximal weighted degree in $\mathcal{G}$, $m_w := \sum_{S(v) \neq S(u)} a_{vu}$ be the summation of weights on inter links. With these notations, we show how the discrepancy between $\mu_0$ and $\mu_1$ changes after conducting feature smoothing for one time over the graph.

**Theorem 4.1.** For an arbitrary graph with nonnegative link weights, after conducting one mean-aggregation over the graph, the consequent representation discrepancy between two sensitive groups $\Delta_{\mathrm{DP}}^{\mathrm{Aggr}} := \|\mathbb{E}_{v \sim U}[\mathrm{Agg}(v) \mid v \in S_0] - \mathbb{E}_{v \sim U}[\mathrm{Agg}(v) \mid v \in S_1]\|_2$ is bounded by

$$\max\{\alpha_{\min}\|\mu_0 - \mu_1\|_\infty - 2\sigma, 0\} \leq \Delta_{\mathrm{DP}}^{\mathrm{Aggr}} \leq \alpha_{\max}\|\mu_0 - \mu_1\|_2 + 2\sqrt{M}\sigma, \tag{3}$$

where $\alpha_{\min} = \min\{\alpha_1, \alpha_2\}, \alpha_{\max} = \max\{\alpha_1, \alpha_2\}, \alpha_1 = |1 - \frac{m_w}{D_{\max}}(\frac{1}{|S_0|} + \frac{1}{|S_1|})|, \alpha_2 = |1 - \frac{|\widetilde{S_0}|}{|S_0|} - \frac{|\widetilde{S_1}|}{|S_1|}|$.

**Remark 2.** Theorem 4.1 shows the lower and upper bound given by the graph structure and the maximum deviation $\sigma$ of vertex representations in each sensitive group on demographic parity after conducting one aggregation function on vertices. The contraction coefficient $\alpha_{\max}$ is a maximum of two absolute terms $\alpha_1$ and $\alpha_2$, where $\alpha_2$ is a constant predetermined by the graph connections. It is worth pointing out that although in the worst case $\alpha_{\max}$ could be 1, e.g., in a complete bipartite graph, in most practical graphs it is strictly less than 1, hence the above upper bound corresponds to a contraction lemma but under some additional error introduced by deviation $\sigma$. We also provide several illustrative graph diagrams for the contraction of this theorem in Appendix B.

To approximate demographic parity after feature smoothing, Theorem 4.1 inspires a strategy to regulate the weights on graph connections to change $\alpha_1$ in $\alpha_{\max}$ so as to minimize the upper bound

---

**Algorithm 1:** Algorithmic routine for **FairAdj**

---

**Input:** vertex features $X$, adjacency matrix $A$, GNNs parameters $\theta$, learning rates $\eta_\theta$ and $\eta_{\widetilde{A}}$

Normalize adjacency matrix $\widetilde{A} \leftarrow D^{-1}A$

Fix the elements with zero in $\widetilde{A}$ and select the non-zero elements for optimization

**while** $\theta$ or $\widetilde{A}$ has not converged **do**

    **for** $t = 1$ **to** $T_1 \triangleright$ `optimize for utility`

    **do**

        Compute $\mathcal{L}_{\text{util}}$ by Eq. (5),    $g_\theta \leftarrow \nabla_\theta \mathcal{L}_{\text{util}}$,    $\theta \leftarrow \theta + \eta_\theta \cdot \text{Adam}(\theta, g_\theta)$

    **for** $t = 1$ **to** $T_2 \triangleright$ `optimize for fairness`

    **do**

        $Z \leftarrow \text{GNN}_\theta(X, \widetilde{A})$,    $\hat{A} \leftarrow ZZ^\top \triangleright$ `reconstruct graph connections`

        Compute $\mathcal{L}_{\text{fair}}$ by Eq. (6),    $g_{\widetilde{A}} \leftarrow \nabla_{\widetilde{A}} \mathcal{L}_{\text{fair}}$

        **for** $v = 1$ **to** $N \triangleright$ `projected gradient descent`

        **do**

            Sort $n$ non-zero elements in $[\widetilde{A} - \eta_{\widetilde{A}} g_{\widetilde{A}}]_{v,*}$ in descending order: $e_1 \geq e_2 \geq \cdots e_n$

            $\gamma \leftarrow \sum_{j=1}^n \mathbf{1}(e_j + \frac{1}{j}(1 - \sum_{i=1}^j e_i) \geq 0) \triangleright \mathbf{1}(\cdot)$: `the indicator function`

            $\beta \leftarrow \frac{1}{\rho}(1 - \sum_{i=1}^\gamma e_i)$

            **for** $u = 1$ **to** $n \triangleright$ `update` $\widetilde{A}$

            **do**

                $[\widetilde{A}]_{v,u} \leftarrow \max\{[\widetilde{A} - \eta_{\widetilde{A}} g_{\widetilde{A}}]_{v,u} + \beta, \, 0\}$

---

**Output:** Link predictive score between vertex $v$ and $u \leftarrow \text{sigmoid}(\text{GNN}_\theta(v, \widetilde{A})^\top \text{GNN}_\theta(u, \widetilde{A}))$

---

for one-layer mean-aggregation. For term $\alpha_1$, obviously if the summation of inter weights are too small ($m_w \to 0$) or too large ($m_w \to D_w \cdot \min\{|S_0|, |S_1|\}$), $\alpha_1$ will approximate to 1. This indicates that increasing the weights on inter links cannot always guarantee to achieve a better demographic parity although inter group connections are always the minority in links, but should regulate this part to a proper range depending on the size of sensitive groups and the connected situation of a graph.

We combine the upper bound with the second feature embedding phase. Here we denote $\mu'_i := \mathbb{E}_{v \sim U}[\text{GNN}_\theta(v, \widetilde{A}) \mid v \in S_i]$, $i = 0, 1$, $Q' := \sup\{\|\text{GNN}_\theta(v, \widetilde{A})\|_2 \mid v \in \mathcal{V}\}$.

**Corollary 4.1.** For $\Delta_{\text{DP}}$ on vertices after passing one layer $\text{GNN}_\theta(X, \widetilde{A}) = \rho(\widetilde{A}XW_\theta)$, we have:

$$\Delta_{\text{DP}} \leq Q' \|\Sigma\|_2 \cdot \|\mu'_0 - \mu'_1\|_2 \leq QL^2 \|\Sigma\|_2 \|W_\theta\|_2^2 \cdot (\alpha\|\mu_0 - \mu_1\|_2 + 2\sqrt{M}\sigma), \qquad (4)$$

where $L$ is the Lipschitz constant for $\rho$. The first inequality holds by Proposition 4.1, the second one is by Theorem 4.1 and the definition of spectral norm and Lipschitz constant, also realizing that $Q' \leq L\|W_\theta\|_2 Q$. Multiple layers of GNNs can be reasoned out similarly.

From the above theorem, we see $\Delta_{\text{DP}}$ can be processed with a tighter upper bound by regulating the weights on edges, and it is also dependent on the property of $W_\theta$, $\rho$, and the error term $O(\sigma)$. When setting $W_\theta$ fixed, our theoretical findings provide a feasible solution that regulating weights on graph connections can achieve better demographic parity on link prediction, and as supplementary, also indicate where or when it cannot perform well with finite layers of GNNs: (1) The solution cannot guarantee arbitrary fairness if we want to preserve the graph structure due to the resistance of $\alpha_1$ in lower bound in Theorem 4.1. (2) When it is already fair enough in the original graph data, which means $\|\mu_0 - \mu_1\|_2$ is small and additional error $O(\sigma)$ is comparable to it, the upper bound cannot be reduced significantly and the solution may not further mitigate the bias. We include a dataset to investigate the potential limitations empirically in response to the above analysis in Appendix D. In the following sections, we implement the inspired algorithm and demonstrate that multiple real-world networks accept this solution with favorable results and a better fairness-utility tradeoff.

## 5   LEARNING FAIR GRAPH CONNECTIONS

The above discussion indicates that when employing GNNs for graph embedding, adjusting the adjacency matrix can assist the model with achieving fairness conditionally. However, searching for the optimal adjacency matrix within hierarchical graph neural networks is a non-trivial problem. In this section, continuing with the preceding analysis, we develop **FairAdj** algorithm to adjust the graph connections and learn a fair adjacency matrix by updating $\widetilde{A}$ while preserving the original graph structure unchanged. In overview, we implement the algorithm by separately optimizing the parameter $W_\theta$ of GNNs towards utility, and adjusting $\widetilde{A}$ towards dyadic fairness by gradient descent and empirical risk minimization with structural and right stochastic constraints. Therefore, **FairAdj** is able to pursue the supplied dyadic fairness and link predictive utility simultaneously.

We employ variational graph autoencoder (Kipf & Welling, 2016b) for feature embedding. A two-layer graph neural network is used as inference model $\text{GNN}_\theta(\cdot, \cdot)$. $Z$ denotes the embedded representations, and dot product between embedded representations is the generative model $p(\cdot)$. The KL-divergence term $KL[\cdot\|\cdot]$ punishes the discrepancy between latent distribution and a Gaussian prior. The objective function to reconstruct graph connections from latent variable can be written as:

$$\max_\theta \quad \mathcal{L}_{\text{util}} := \mathbb{E}_{\text{GNN}_\theta(Z|X,\widetilde{A})}[\log p(A \mid Z)] - KL[\text{GNN}_\theta(Z \mid X, \widetilde{A})\|\mathcal{N}(0,1)]. \tag{5}$$

For fairness, we impel $\mathcal{L}_{\text{fair}}$ to empirically seek for better graph connections, then update $\widetilde{A}$ with constraints. Specifically, we optimize the normalized adjacency matrix $\widetilde{A}$ as follows:

$$\min_{\widetilde{A}} \quad \mathcal{L}_{\text{fair}} := \|\mathbb{E}_{v,u\sim U\times U}[\hat{a}_{vu} \mid S(v) = S(u)] - \mathbb{E}_{v,u\sim U\times U}[\hat{a}_{vu} \mid S(v) \neq S(u)]\|^2,$$
$$\text{s.t.} \quad (1). \, [\widetilde{A}]_{vu} = 0, \text{ if } [A]_{vu} = 0, \quad (2). \, \widetilde{A}\mathbb{1} = \mathbb{1}, \, \widetilde{A} \geq 0, \tag{6}$$

where $\hat{a}_{vu}$ takes value in $\hat{A} = ZZ^\top$ and $\mathbb{1}$ is the all-one vector with size $N$. The two constraints are necessary for optimizing $\widetilde{A}$: (1) Elements with zero value should be maintained, meaning no new links can be established during optimization. This restriction is proposed to preserve utility, due to adding fictitious links might mislead the directions of message passing, and further corrupt the representation learning. Therefore, we only adapt weights on existing edges and preserve the original graph structure. (2) In consistent with the initial left normalization $\widetilde{A} = D^{-1}A$, the optimized matrix should still remain a right stochastic matrix. This can restrict the largest eigenvalue of adjacency matrix $\|\widetilde{A}\|_2 = 1$, hence avoid numerical instabilities and exploding gradients when training $W_\theta$ towards $\mathcal{L}_{\text{util}}$. In practice, we observe explosions during training with no constraints applied on $\widetilde{A}$.

For the first constraint in Eq. (6), we only compute gradients and update the elements in $\widetilde{A}$ which have non-zero initialization. For the second one, after selecting the variable to optimize, we employ projected gradient descent by (Wang & Carreira-Perpinán, 2013) to satisfy the constraint while minimizing $\mathcal{L}_{\text{fair}}$. Once given computed gradients on $\widetilde{A}$ denoted as $\nabla_{\widetilde{A}}\mathcal{L}_{\text{fair}}$, with the corresponding learning rate $\eta_{\widetilde{A}}$, we have the following optimization problem that update $\widetilde{A}$ by projecting $\widetilde{A} - \eta_{\widetilde{A}}\nabla_{\widetilde{A}}\mathcal{L}_{\text{fair}}$ into the feasible region with the minimum Euclidean distance:

$$\min_{\widetilde{A} - (\eta_{\widetilde{A}}\nabla_{\widetilde{A}}\mathcal{L}_{\text{fair}})'} \quad \sum_v \|[\widetilde{A} - \eta_{\widetilde{A}}\nabla_{\widetilde{A}}\mathcal{L}_{\text{fair}}]_{v,*} - [\widetilde{A} - (\eta_{\widetilde{A}}\nabla_{\widetilde{A}}\mathcal{L}_{\text{fair}})']_{v,*}\|^2$$
$$\text{s.t.} \quad (\widetilde{A} - (\eta_{\widetilde{A}}\nabla_{\widetilde{A}}\mathcal{L}_{\text{fair}})')\mathbb{1} = \mathbb{1} \text{ and } \widetilde{A} - (\eta_{\widetilde{A}}\nabla_{\widetilde{A}}\mathcal{L}_{\text{fair}})' \geq 0, \tag{7}$$

where $[\widetilde{A}]_{v,*}$ are the row-wise elements for $\widetilde{A}$, namely, all the connections on $v$. $\widetilde{A} - (\eta_{\widetilde{A}}\nabla_{\widetilde{A}}\mathcal{L}_{\text{fair}})'$ is the update for $\widetilde{A}$ after projection. Since $\widetilde{A}$ is row-wise independent and the objective function is strictly convex for this quadratic program, there exists a unique solution for each row. Solution details for projected gradient descent are restated as a part of our algorithmic pipeline.

The algorithmic routine is elaborated in Algorithm 1. $\theta$ and $\widetilde{A}$ are optimized iteratively with $T_1$ and $T_2$ epochs for co-adaptation. Compared to adversarial training method on graph embedding (Bose & Hamilton, 2019b), which requires a hyperparameter to control the fairness-utility tradeoff, we also find regulating the convergence of $\widetilde{A}$ has a similar effect as well. That is because the more $\widetilde{A}$ changes, the further $\widetilde{A}$ is away from the original graph connections, and consequently, the more

Table 1: Statistic for datasets in experiments.

| Dataset | # Vertex | # Edge | # Class | # Intra | # Inter | Intra Ratio | Inter Ratio | Dis. Ratio |
|---|---|---|---|---|---|---|---|---|
| *Oklahoma97* | 3,111 | 73,230 | 2 | 46,368 | 26,862 | 1.92e-2 | 1.11e-2 | 1.73 |
| *UNC28* | 4,018 | 65,287 | 2 | 36,212 | 29,075 | 8.76e-3 | 7.38e-3 | 1.19 |
| *Facebook#1684* | 786 | 14,024 | 2 | 7,989 | 6,035 | 4.76e-2 | 4.30e-2 | 1.11 |
| *Cora* | 2,708 | 5,278 | 7 | 4,275 | 1,003 | 6.51e-3 | 3.30e-4 | 19.73 |
| *Citeseer* | 3,312 | 4,660 | 6 | 2,089 | 2,571 | 2.13e-3 | 5.70e-4 | 3.74 |
| *Pubmed* | 19,717 | 44,327 | 3 | 33,443 | 10,884 | 4.80e-4 | 9.00e-5 | 5.33 |

Table 2: Experimental results on *UNC28*.

| Method | AUC ↑ | AP ↑ | $\Delta_{DP}$ ↓ | $\Delta_{true}$ ↓ | $\Delta_{false}$ ↓ | $\Delta_{FNR}$ ↓ | $\Delta_{TNR}$ ↓ |
|---|---|---|---|---|---|---|---|
| VGAE | **87.63** $\pm$ 0.56 | **88.69** $\pm$ 0.65 | 2.24 $\pm$ 0.42 | 1.50 $\pm$ 0.41 | 0.44 $\pm$ 0.36 | 7.62 $\pm$ 0.84 | 2.18 $\pm$ 0.72 |
| node2vec | 87.22 $\pm$ 0.30 | 87.10 $\pm$ 0.37 | 2.75 $\pm$ 0.78 | 1.30 $\pm$ 0.53 | 1.05 $\pm$ 0.93 | 12.56 $\pm$ 1.12 | 2.24 $\pm$ 0.92 |
| Fairwalk | 87.18 $\pm$ 0.30 | 87.07 $\pm$ 0.37 | 2.79 $\pm$ 0.70 | 1.17 $\pm$ 0.49 | 0.90 $\pm$ 0.92 | 12.71 $\pm$ 1.11 | 2.20 $\pm$ 0.96 |
| FairAdj$_{T2=5}$ | 86.98 $\pm$ 0.54 | 87.75 $\pm$ 0.65 | **1.53** $\pm$ 0.35 | **0.32** $\pm$ 0.29 | **0.41** $\pm$ 0.35 | 2.84 $\pm$ 0.74 | 2.22 $\pm$ 0.68 |
| FairAdj$_{T2=20}$ | 87.04 $\pm$ 0.55 | 87.80 $\pm$ 0.65 | 1.57 $\pm$ 0.36 | 0.34 $\pm$ 0.31 | 0.42 $\pm$ 0.35 | **2.76** $\pm$ 0.75 | **2.16** $\pm$ 0.73 |

damage in utility but more enhancement in fairness. Thanks to this favorable property, in experiments, comparing to adversarially remove sensitive attributes, we present multiple options for $T_2$ to control the convergence and observe a more favorable fairness-utility tradeoff shown by our method.

# 6 EXPERIMENTS

We present empirical analysis on six real-world datasets, compared with baseline methods in terms of seven evaluative metrics on both fairness and utility. Approaching to applications, we testify that our method can enhance the diversity in recommendations. Due to the space limitation, we only showcase partial results but defer the rest in Appendix D. Moreover, as an intermediate result, we demonstrate vertex representations are embedded more fairly assessed by fair clustering in Appendix E.

## 6.1 SETTINGS

**Datasets.** We conduct experiments on real-world social networks and citation networks including *Oklahoma97*, *UNC28* (Traud et al., 2011), *Facebook#1684*, *Cora*, *Citeseer*, and *Pubmed*. *Oklahoma97* and *UNC28* are two school social networks. A link represents a friendship relation in social media, and every user has a profile for vertex features, including student/faculty status, gender (sensitive attribute), major, etc. *Facebook#1684* is a social ego network from Facebook app. As the rest three citation networks, each vertex represents an article with bag-of-words descriptions as features. A link stands for a citation regardless the direction. We set the category of an article as the sensitive attribute. Statistic for datasets are summarized in Table 1, where #Class is the number of sensitive groups, #Intra/Inter Ratio represents the ratio that the number of actual intra/inter links v.s. the number of links if the graph is fully connected. These two terms show the density of intra/inter links. Dis. Ratio (abbreviated from disparity ratio) is calculated by divide intra ratio into inter ratio. Dis. Ratio equaling to one implies intra/inter connections are perfectly balanced in existing graph, and the degree of deviation from 1 indicates how skew the link connections are.

**Baselines and Protocols.** We involve four baseline methods. Variational graph autoencoder (VGAE) (Kipf & Welling, 2016b) inherits from variational autoencoder, which uses two GNN-layer as the inference model and leverages latent variables to reconstruct the graph connections. Node2vec (Grover & Leskovec, 2016) is a widely used graph embedding approach based on random walk. Fairwalk (Rahman et al., 2019) is built upon node2vec and designed specifically for fairness issues. It modifies the transition probability for one vertex according to the sensitive attribute of its neighbors. The last one is adversarial training on vertex representations (Bose & Hamilton, 2019b), which aims to minimize the discrepancy between different sensitive groups by optimizing parameters in GNN. Besides the standard pipeline for utility, it additionally trains the networks to confuse a discriminator, meanwhile training the discriminator to distinguish the embedded features with different sensitive attributes. A hyperparameter $\lambda$ is used in the overall objective function to balance

Table 3: Experimental results on *Citeseer*.

| Method | AUC ↑ | AP ↑ | $\Delta_{DP}$ ↓ | $\Delta_{true}$ ↓ | $\Delta_{false}$ ↓ | $\Delta_{FNR}$ ↓ | $\Delta_{TNR}$ ↓ |
|---|---|---|---|---|---|---|---|
| VGAE | **81.77** ± 1.23 | **85.57** ± 1.39 | 11.24 ± 1.83 | 3.37 ± 2.33 | 2.14 ± 1.26 | 10.81 ± 3.61 | 11.31 ± 2.99 |
| node2vec | 81.21 ± 1.35 | 84.69 ± 1.26 | 14.49 ± 3.38 | 4.02 ± 2.74 | 6.82 ± 4.62 | 7.37 ± 3.07 | 12.50 ± 4.21 |
| Fairwalk | 81.69 ± 1.50 | 84.97 ± 1.23 | 13.50 ± 2.97 | 3.30 ± 2.49 | 5.33 ± 3.83 | **7.26** ± 3.30 | 11.34 ± 3.19 |
| FairAdj$_{T2=2}$ | 80.45 ± 1.34 | 84.47 ± 1.43 | 9.57 ± 1.84 | 2.55 ± 2.02 | 1.70 ± 1.47 | 9.87 ± 3.17 | 10.27 ± 3.23 |
| FairAdj$_{T2=20}$ | 78.84 ± 1.38 | 82.74 ± 1.46 | **7.81** ± 1.80 | **1.85** ± 1.66 | **1.30** ± 1.41 | 9.22 ± 2.95 | **10.01** ± 3.13 |

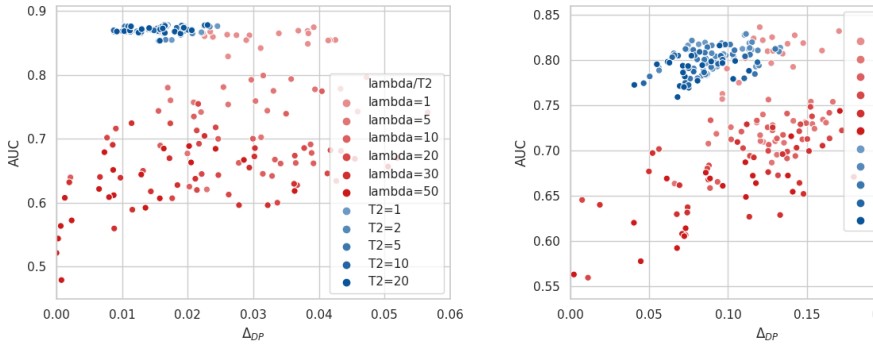

Figure 1: Comparison with adversarial training method (Bose & Hamilton, 2019b) in terms of the tradeoff between utility and fairness. Left: *UNC28*; Right: *Citeseer*. Blue points denote **FairAdj** with different $T_2$ values, red points represent (Bose & Hamilton, 2019b) with different $\lambda$ values.

the tradeoff between utility and fairness. We vary $\lambda$ in experiments and make comparisons to various results given by adversarial training. For all experiments, we randomly remove 10% links from the graph and reserve them for evaluation, and equivalently, the same number of false links are sampled in the evaluation phase. For one dataset, we repeat experiments with different train/test splits for 20 times. Full experimental configurations are available in Appendix C.

**Metrics.** We evaluate the utility of link prediction using Area Under the Curve (AUC) and Average Precision (AP). Fairness is evaluated towards $\Delta_{DP}$, as well as the disparity on the expected score on all the true samples $\Delta_{true}$ and false samples $\Delta_{false}$. Besides these, following the suggested fairness notions (Hardt et al., 2016), we compute the maximum gap of true negative rate (TNR) and false negative rate (FNR). Illustratively, let the conditional cumulative distribution function of score $R$ be evaluated by a threshold $\tau$ with a given label written as $F_y^s(\tau) := \Pr(R \leq \tau | Y = y, S = s)$, $y \in \{0, 1\}$, $s \in \{\text{intra}, \text{inter}\}$. With these notations, the maximum gap in true negative rate can be expressed as $\Delta_{TNR} := \max_\tau |F_0^{\text{intra}}(\tau) - F_0^{\text{inter}}(\tau)|$, and similar for false positive rate $\Delta_{FNR} := \max_\tau |F_1^{\text{intra}}(\tau) - F_1^{\text{inter}}(\tau)|$. These two terms also reflect the disparity in true positive rate (TPR) and false positive rate (FPR) due to TPR $= 1 - $ FNR and FPR $= 1 - $ TNR.

## 6.2 RESULTS AND ANALYSIS

Table 2 and 3 list quantitative results on *UNC28* and *Citeseer* comparing to VGAE, node2vec, and Fairwalk. Two choices of $T_2$ are presented here, where $T_2$ in a smaller value pursues a comparable performance in utility (a little bit lower in AUC but higher in AP) to random walk based methods, and at the same time performs much better in fairness. We also present $T_2 = 20$ since we observe a convergence on the adjacency matrix. The results indicate that **FairAdj** achieves the best on various statistics for dyadic fairness and with only a small sacrifice in predictive utility.

**A Better Tradeoff.** Figure 1 plots every experimental result for our method and adversarial training with various fairness-utility regulated hyperparameters. Two observations explain why our method surpasses the adversarial technique. (1) Blue dots are closer to the left top corner than the red on the whole, meaning the same level of fairness is achieved with less sacrifice in utility. An explanation for this favorable property is that, the adversarial training method neglects the graph connections and

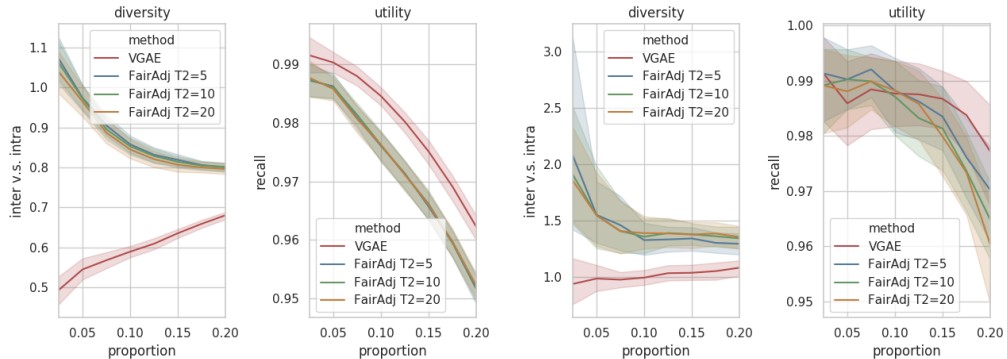

Figure 2: Diversity and utility in recommendations. Left: *UNC28*; Right: *Citeseer*. X-axis 'proportion' means we investigate the top x% valued links and check the ratio between inter and intra links that presented in y-axis.

only diminish the group discrepancy, where two irrelevant instances with no connection but from different groups may be closely mapped, thus greatly damage the utility. The optimization on $\widetilde{A}$ does facilitate the feature smoothing across groups which is not indicated in the original adjacency matrix, but still considers the graph connections. (2) Additionally, blue dots are more aggregated, suggesting our methods escape from the instability of min-max optimization and acting more robust to different train/test splits. However, as shown, **FairAdj** cannot achieve arbitrary small in $\Delta_{\mathrm{DP}}$ as red dots do. This is indicated in Section 4 as the first potential limitation.

**Diversity in Recommendations.**   We examine the top-scored links in evaluation at a certain proportion in terms of its diversity and utility, shown in Figure 2. This exploration can be useful when conducting recommendations according to scores in descending order. For a fixed proportion, we report the diversity as the number of inter links divided by the number of intra links, and the utility as the recall rate among these recommendations. Figures show that as a direct product by dyadic fairness, **FairAdj** enhances the diversity in recommendations but is achieved at a sacrifice of utility.

## 7   CONCLUSION

We studied the dyadic fairness in graph-structured data. We theoretically analyzed how the connections in graph links affect dyadic fairness of demographic parity when employing graph neural networks for representation learning. On the basis of the foregoing analysis, we proposed **FairAdj** to learn a fair adjacency matrix, and pursued the dyadic fairness and prediction utility simultaneously. Empirical validations demonstrated the achievement of fairness and a better fairness-utility tradeoff.

## ACKNOWLEDGEMENT

We would like to thank Zizhang Chen and Wei Lu for the helpful discussions, and Lizi Liao for providing the Oklahoma97/UNC28 datasets. This work is partially supported by NSF OAC 1920147.

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

## A  PROOF

**Proposition 4.1.** For a link prediction function $g(\cdot, \cdot)$ modeled as inner product $g(v, u) = v^\top \Sigma u$, where $\Sigma \in \mathbb{S}_{++}^M$ is a positive-definite matrix, $\exists Q > 0, \forall v \sim \mathcal{V}, \|v\|_2 \le Q$, for $\mathbb{E}_{v \sim U}[v] \in \mathbb{R}^M$, for dyadic fairness based on demographic parity, if $\|\mathbb{E}_{v \sim U}[v \mid v \in S_0] - \mathbb{E}_{v \sim U}[v \mid v \in S_1]\|_2 \le \delta$,

$$\Delta_{\text{DP}} := |\mathbb{E}_{(v,u) \sim U \times U}[g(v, u) \mid S(v) = S(u)] - \mathbb{E}_{(v,u) \sim U \times U}[g(v, u) \mid S(v) \ne S(u)]| \le Q\|\Sigma\|_2 \cdot \delta. \tag{2}$$

*Proof.* To simplify the notations, we use $p := \mathbb{E}_{v \sim U}[v \mid v \in S_0] \in \mathbb{R}^M$ and $q := \mathbb{E}_{v \sim U}[v \mid v \in S_1] \in \mathbb{R}^M$ to denote the expectations in representations for $S_0$ and $S_1$ respectively.

$$\begin{aligned}
|\mathbb{E}_{\text{intra}} - \mathbb{E}_{\text{inter}}| &= \left| \mathbb{E}[v^\top \Sigma u \mid v \in S_0, u \in S_1] - \mathbb{E}[v^\top \Sigma u \mid v \in S_0, u \in S_0 \vee v \in S_1, u \in S_1] \right| \\
&= \left| p^\top \Sigma q - \left( \frac{|S_0|^2}{|S_0|^2 + |S_1|^2} p^\top \Sigma p + \frac{|S_1|^2}{|S_0|^2 + |S_1|^2} q^\top \Sigma q \right) \right| \\
&= \left| (q - p)^\top \left( \frac{|S_0|^2}{|S_0|^2 + |S_1|^2} \Sigma p - \frac{|S_1|^2}{|S_0|^2 + |S_1|^2} \Sigma q \right) \right|
\end{aligned}$$

To simplify the notation, we will use $\alpha := |S_0|^2/(|S_0|^2 + |S_1|^2)$ and $\beta := |S_1|^2/(|S_0|^2 + |S_1|^2)$

$$\begin{aligned}
&\le \|q - p\|_2 \cdot \|\alpha \Sigma p - \beta \Sigma q\|_2 \\
&\le \delta \cdot \|\Sigma\|_2 \cdot (\|\alpha p\|_2 + \|\beta q\|_2) \\
&= Q\|\Sigma\|_2 \cdot \delta,
\end{aligned}$$

which completes the proof. The first inequality above is due to Cauchy-Schwarz, and the second one is by the definition of spectral norm. The last equality holds by the linearity of expectation: if $\forall v \in \mathcal{V}, \|v\|_2 \le Q$, then $\|\mathbb{E}[v]\|_2 \le \mathbb{E}[\|v\|_2] \le Q$. ∎

**Theorem 4.1.** For an arbitrary graph with nonnegative link weights, after conducting one mean-aggregation over the graph, the consequent representation discrepancy between two sensitive groups $\Delta_{\text{DP}}^{\text{Aggr}} := \|\mathbb{E}_{v \sim U}[\text{Agg}(v) \mid v \in S_0] - \mathbb{E}_{v \sim U}[\text{Agg}(v) \mid v \in S_1]\|_2$ is bounded by

$$\max\{\alpha_{\min}\|\mu_0 - \mu_1\|_\infty - 2\sigma, 0\} \le \Delta_{\text{DP}}^{\text{Aggr}} \le \alpha_{\max}\|\mu_0 - \mu_1\|_2 + 2\sqrt{M}\sigma, \tag{3}$$

where $\alpha_{\min} = \min\{\alpha_1, \alpha_2\}, \alpha_{\max} = \max\{\alpha_1, \alpha_2\}, \alpha_1 = |1 - \frac{m_w}{D_{\max}}(\frac{1}{|S_0|} + \frac{1}{|S_1|})|, \alpha_2 = |1 - \frac{|\widetilde{S_0}|}{|S_0|} - \frac{|\widetilde{S_1}|}{|S_1|}|$.

*Proof.* The feature representation of $v$ after conducting one mean-aggregation is

$$\text{Agg}(v) = \frac{1}{\deg_w(v)} \sum_{u \in \Gamma(u)} a_{vu} u = \frac{1}{\deg_w(v)} \left( \sum_{u \in \Gamma(u) \cap S_0} a_{vu} u + \sum_{u \in \Gamma(u) \cap S_1} a_{vu} u \right).$$

Here we separate the summation of neighbor features into two parts in terms of the sensitive attribute.

We use the bracket notation to abbreviate the range of a vector. That is, if a vector $u$ satisfies $\mu - \sigma \le u \le \mu + \sigma$, we abbreviate this as $u \in [\mu \pm \sigma]$.

Consider the unilateral case $v \in S_0$, we have

$$\text{Agg}(v) \in \left[ \frac{\sum_{u \in \Gamma(v) \cap S_0} a_{vu} \mu_0}{\deg_w(v)} + \frac{\sum_{u \in \Gamma(v) \cap S_1} a_{vu} \mu_1}{\deg_w(v)} \pm \sigma \cdot \mathbb{1} \right]$$

$$\in \left[ \left( \mu_0 + \frac{\sum_{u \in \Gamma(v) \cap S_1} a_{vu}}{\deg_w(v)} (\mu_1 - \mu_0) \right) \pm \sigma \cdot \mathbb{1} \right]$$

where $\mathbb{1}$ is the all-one vector with proper size.

The first derivation is due to the fact that each $u \in S_0$ lies in the range of $[\mu_0 \pm \sigma \cdot \mathbb{1}]$ and each $u \in S_1$ lies in the range of $[\mu_1 \pm \sigma \cdot \mathbb{1}]$. The second one is by the definition of weighted degree.

Using $\beta_v = \sum_{v \in \Gamma(v) \cap S_{\text{opp}(v)}} a_{vu}/\deg_w(v)$ where $S_{\text{opp}(v)}$ is the opposite sensitive group where $v$ belongs. The expectation of $\text{Agg}(v)$ for $S_0$ is

$$\mathbb{E}_{v \sim U}[\text{Agg}(v) \mid v \in S_0] \in [(\frac{1}{|S_0|} \sum_{v \in S_0} (\mu_0 + \beta_v(\mu_1 - \mu_0))) \pm \sigma \cdot \mathbb{1}]$$

$$\in [(\mu_0 + \frac{1}{|S_0|} \sum_{v \in S_0} \beta_v(\mu_1 - \mu_0)) \pm \sigma \cdot \mathbb{1}].$$

And for $v \in S_1$ we have

$$\mathbb{E}_{v \sim U}[\text{Agg}(v) \mid v \in S_1] \in [(\mu_1 + \frac{1}{|S_1|} \sum_{v \in S_1} \beta_v(\mu_0 - \mu_1)) \pm \sigma \cdot \mathbb{1}].$$

Based on the above two terms, the gap in expectation of two groups after passing one mean-aggregation layer becomes

$$\mathbb{E}_{v \sim U}[\text{Agg}(v) \mid v \in S_0] - \mathbb{E}_{v \sim U}[\text{Agg}(v) \mid v \in S_1] \in [(1 - (\frac{1}{|S_0|} \sum_{v \in S_0} \beta_v + \frac{1}{|S_1|} \sum_{v \in S_1} \beta_v)) \cdot (\mu_0 - \mu_1) + 2\sigma \cdot \mathbb{1}].$$

Next we study the range of $\alpha' := 1 - (|S_0|^{-1} \sum_{v \in S_0} \beta_v + |S_1|^{-1} \sum_{v \in S_1} \beta_v)$. First we consider the term $|S_0|^{-1} \sum_{v \in S_0} \beta_v$. Since $\deg_w(v) \leq D_{\max}, \forall v \in \mathcal{V}$, we have

$$\sum_{v \in S_0} \beta_v = \sum_{v \in S_0} \frac{\sum_{u \in \Gamma(v) \cap S_1} a_{vu}}{\deg_w(v)} \geq \frac{1}{D_{\max}} \sum_{v \in S_0} \sum_{u \in \Gamma(v) \cap S_1} a_{vu} = \frac{m_w}{D_{\max}}.$$

For non-negative weights,

$$D_{\max} \geq \deg_w(v) = \sum_{u \in \Gamma(v) \cap S_0} a_{vu} + \sum_{u \in \Gamma(v) \cap S_1} a_{vu} \geq \sum_{u \in \Gamma(v) \cap S_1} a_{vu}.$$

This means for $v \in S_0$,

$$\beta_v = \frac{\sum_{u \in \Gamma(u) \cap S_1} a_{vu}}{\deg_w(u)} \leq 1,$$

thus,

$$\sum_{v \in S_0} \beta_v = \sum_{v \in \widetilde{S_0}} \beta_v \leq |\widetilde{S_0}|.$$

The first equality holds because $\beta_v = 0$ when $v \in S_0/\widetilde{S_0}$, meaning $v$ doesn't contain any inter-edges.

Since the analysis for $S_1$ is similar, we derive the lower and upper bounds for $|S_i|^{-1} \sum_{v \in S_i} \beta_v$, $i = 0, 1$

$$\frac{1}{|S_i|} \cdot \frac{m_w}{D_{\max}} \leq \frac{1}{|S_i|} \sum_{v \in S_i} \beta_v \leq (\frac{|\widetilde{S_i}|}{|S_i|}), \quad i = 0, 1.$$

Based on the above results, we give the bound for $\alpha'$ as follows:

$$\alpha' \in [\, 1 - (\frac{|\widetilde{S_0}|}{|S_0|} + \frac{|\widetilde{S_1}|}{|S_1|}), \, 1 - \frac{m_w}{D_{\max}}(\frac{1}{|S_0|} + \frac{1}{|S_1|}) \,],$$

Let $\alpha_{\min}$ and $\alpha_{\max}$ be lower bound and upper bower of $|\alpha'|$, we have

$$\alpha_{\max} = \max\{1 - (\frac{|\widetilde{S_0}|}{|S_0|} + \frac{|\widetilde{S_1}|}{|S_1|}), \, 1 - \frac{m_w}{D_{\max}}(\frac{1}{|S_0|} + \frac{1}{|S_1|})\}$$

$$\alpha_{\min} = \min\{1 - (\frac{|\widetilde{S_0}|}{|S_0|} + \frac{|\widetilde{S_1}|}{|S_1|}), \, 1 - \frac{m_w}{D_{\max}}(\frac{1}{|S_0|} + \frac{1}{|S_1|})\}$$

Thus we give the upper bound of $\Delta_{\mathrm{DP}}^{\mathrm{Aggr}}$:

$$\Delta_{\mathrm{DP}}^{\mathrm{Aggr}} \leq \alpha_{\max}\|\mu_0 - \mu_1\|_2 + 2\sqrt{M}\sigma \tag{8}$$

where the second part in RHS is due to $2\sigma \cdot \|\mathbb{1}\|_2 = 2\sqrt{M}\sigma$.

Next we consider $i$-th entrance of $\mu_0$ and $\mu_1$, denoted as $\mu_0^i$ and $\mu_1^i$ respectively. The $i$-the entrance of $\mathbb{E}_{v \sim U}[\mathrm{Agg}(v) \mid v \in S_0] - \mathbb{E}_{v \sim U}[\mathrm{Agg}(v) \mid v \in S_1]$ take nonzero values if and only if

$$|(1 - (\frac{1}{|S_0|}\sum_{v \in S_0}\beta_v + \frac{1}{|S_1|}\sum_{v \in S_1}\beta_v)) \cdot (\mu_0^i - \mu_1^i)| \geq 2\sigma$$

Thus we obtain the lower bound of $\Delta_{\mathrm{DP}}^{\mathrm{Aggr}}$:

$$\Delta_{\mathrm{DP}}^{\mathrm{Aggr}} \geq \max\{\alpha_{\min}\|\mu_0 - \mu_1\|_\infty - 2\sigma, 0\} \tag{9}$$

which completes the proof. ■

## B    COMPLEMENTARY DIAGRAMS TO THEOREM 4.1

We provide diagrams to help better understand the upper bound in Theorem 4.1.

Figure 3 provides a common case that the gap in expectation between two sensitive groups shrinks after mean-aggregation. Here the maximal deviation term $\sigma$ can be neglected since it is much smaller than the expectation gap.

Figure 4 provides a case that the term $\sigma$ is not negligible against the expectation gap between two sensitive groups. Here $\sigma = 100$ and the gap equals to 0. After aggregation, we see the new expectation gap becomes 20, showing that the discrepancy in representations increases.

Figure 5 provides another case that the contraction coefficient $\alpha$ equals to 1 due to the resistance of $\alpha_2$. Here all vertices possess inter links, and the graph is a complete bipartite graph. Then the aggregation fully exchanges the sensitive information, and thus the representation discrepancy remains unchanged.

Cases in Figure 4 and 5 are also pointed out by the analysis in Section 4.

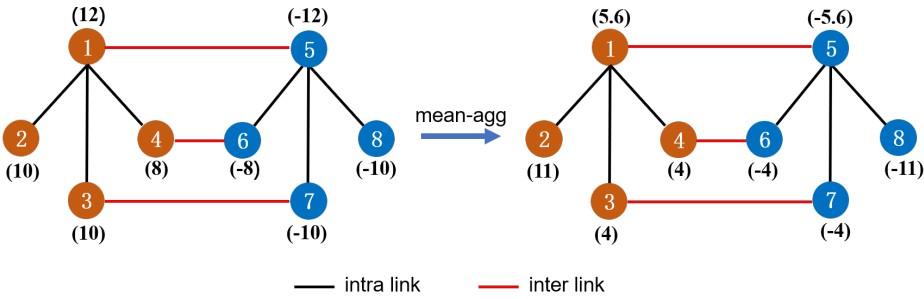

Figure 3: An illustrative graph example with two protected groups $S_0$ and $S_1$. All vertices have self-loop. The expectation gap shrinks after mean aggregation. Here, $|\mathbb{E}_{v \sim U}[v|v \in S_0] - \mathbb{E}_{v \sim U}[v|v \in S_1]| = 20$, $\sigma = 2$ and all link weights are equal. After aggregation, $|\mathbb{E}_{v \sim U}[\mathrm{Agg}(v)|v \in S_0] - \mathbb{E}_{v \sim U}[\mathrm{Agg}(v)|v \in S_1]| = |6.15 - (-6.15)| = 12.3 < 20$.

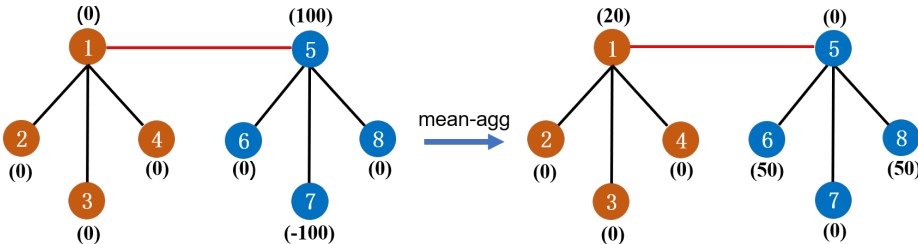

Figure 4: Case 1: The maximal deviation term $O(\sigma)$ is not negligible. Here $\sigma = 100$ and all link weights are equal. All vertices have self-loop. $|\mathbb{E}_{v \sim U}[v|v \in S_0] - \mathbb{E}_{v \sim U}[v|v \in S_1]| = 0$. But after mean-aggregation, $|\mathbb{E}_{v \sim U}[\text{Agg}(v)|v \in S_0] - \mathbb{E}_{v \sim U}[\text{Agg}(v)|v \in S_1]| = |5 - 25| = 20 > 0$.

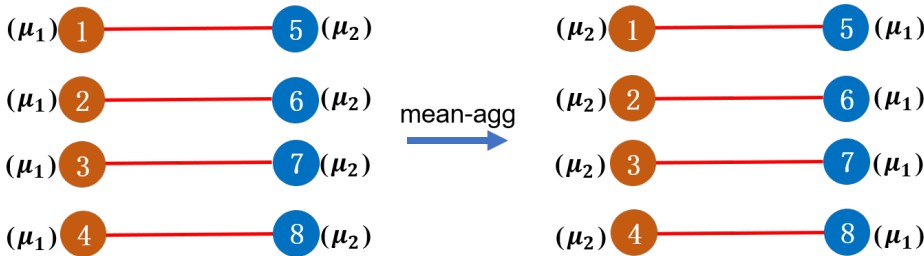

Figure 5: Case 2: The contraction coefficient $\alpha$ equals to 1. This happens when the graph is a complete bipartite graph. Mean-aggregation fully exchanges the sensitive information and the gap of two groups remains unchanged.

## C  EXPERIMENTAL CONFIGURATIONS

For all experiments, we set $T_1 = 50$ and the total epochs which contain $T_1$ and $T_2$ equal to 4. Graph neural networks are applied with two hidden layers with size 32 and 16 respectively. $\eta_\theta$ is set to 0.01. For $\eta_{\tilde{A}}$ for different datasets, we have: *Oklahoma97*: 0.1; *UNC28*: 0.1; *Cora*: 0.2; *Citeseer*: 0.5. Experiments are conducted on Nvidia Titan RTX graphics card.

## D  ADDITIONAL RESULTS

We present experimental results for *Citeseer* and *UNC28* in this section. All the results deliver similar conclusions as we state in the main body of this paper. Additionally, we include another dataset *Facebook#1684* in response to the second limitation as indicated in Section 4. In this case, $\Delta_{\text{DP}}$, $\Delta_{\text{true}}$, $\Delta_{\text{false}}$ are already small as given by VGAE, and FairAdj is not able to further minimize the gap.

Table 4: Experimental results on *Oklahoma97*.

| Method | AUC ↑ | AP ↑ | $\Delta_{\text{DP}}$ ↓ | $\Delta_{\text{true}}$ ↓ | $\Delta_{\text{false}}$ ↓ | $\Delta_{\text{FNR}}$ ↓ | $\Delta_{\text{TNR}}$ ↓ |
|---|---|---|---|---|---|---|---|
| VGAE | **90.13** ± 0.32 | **91.24** ± 0.37 | 8.73 ± 0.38 | 8.56 ± 0.44 | 0.40 ± 0.32 | 36.51 ± 1.41 | 2.26 ± 0.92 |
| node2vec | 86.49 ± 0.35 | 84.09 ± 0.50 | 7.23 ± 0.64 | 3.35 ± 0.45 | 1.08 ± 0.97 | 32.55 ± 1.32 | 2.36 ± 0.69 |
| Fairwalk | 86.56 ± 0.32 | 84.23 ± 0.44 | 7.31 ± 0.62 | 3.49 ± 0.47 | 1.13 ± 0.85 | 32.77 ± 1.20 | 2.18 ± 0.69 |
| FairAdj$_{\text{T2}=5}$ | 84.92 ± 0.81 | 85.07 ± 0.92 | 3.60 ± 0.35 | 0.40 ± 0.32 | 0.33 ± 0.28 | **4.00** ± 0.88 | **2.02** ± 0.76 |
| FairAdj$_{\text{T2}=20}$ | 81.01 ± 1.01 | 80.79 ± 0.93 | **2.96** ± 0.30 | **0.38** ± 0.31 | **0.32** ± 0.25 | 5.61 ± 1.06 | 2.03 ± 0.92 |

Table 5: Experimental results on *Cora*.

| Method | AUC ↑ | AP ↑ | $\Delta_{DP}$ ↓ | $\Delta_{true}$ ↓ | $\Delta_{false}$ ↓ | $\Delta_{FNR}$ ↓ | $\Delta_{TNR}$ ↓ |
|---|---|---|---|---|---|---|---|
| VGAE | **88.48** ± 0.88 | **90.81** ± 0.78 | 26.74 ± 1.51 | 9.99 ± 2.32 | 10.26 ± 1.59 | 28.25 ± 4.46 | 26.71 ± 3.83 |
| node2vec | 87.93 ± 0.75 | 87.82 ± 1.06 | 39.99 ± 2.75 | 6.63 ± 3.58 | 27.86 ± 4.94 | 23.66 ± 4.73 | 32.96 ± 5.24 |
| Fairwalk | 88.04 ± 0.84 | 88.10 ± 1.20 | 40.49 ± 2.58 | 7.30 ± 3.28 | 29.43 ± 4.86 | 23.74 ± 4.19 | 33.79 ± 5.08 |
| FairAdj$_{T2=5}$ | 86.00 ± 1.12 | 88.32 ± 0.86 | 21.05 ± 1.26 | 6.99 ± 2.24 | 6.14 ± 1.59 | 20.72 ± 3.62 | 19.46 ± 3.62 |
| FairAdj$_{T2=20}$ | 83.85 ± 1.07 | 86.08 ± 0.93 | 17.87 ± 1.18 | **5.40** ± 2.23 | **3.74** ± 1.46 | **16.75** ± 4.87 | **15.37** ± 3.84 |

Table 6: Experimental results on *Pubmed*.

| Method | AUC ↑ | AP ↑ | $\Delta_{DP}$ ↓ | $\Delta_{true}$ ↓ | $\Delta_{false}$ ↓ | $\Delta_{FNR}$ ↓ | $\Delta_{TNR}$ ↓ |
|---|---|---|---|---|---|---|---|
| VGAE | **91.20** ± 0.85 | **91.26** ± 0.80 | 20.88 ± 1.48 | 4.19 ± 0.93 | 8.04 ± 1.83 | 12.01 ± 2.92 | 19.18 ± 4.16 |
| node2vec | 74.27 ± 1.23 | 79.24 ± 1.29 | 19.14 ± 0.93 | 3.38 ± 2.57 | 8.90 ± 2.56 | 6.65 ± 2.21 | 10.91 ± 1.88 |
| fairwalk | 73.43 ± 1.11 | 78.96 ± 1.24 | 18.42 ± 1.65 | 3.11 ± 1.84 | 7.79 ± 3.49 | **6.61** ± 2.28 | 10.93 ± 2.54 |
| FairAdj$_{T2=5}$ | 88.64 ± 1.09 | 88.21 ± 1.22 | 16.06 ± 0.98 | 1.96 ± 0.82 | 4.40 ± 1.28 | 8.93 ± 2.90 | 12.75 ± 1.56 |
| FairAdj$_{T2=20}$ | 87.53 ± 1.03 | 87.10 ± 1.17 | **14.73** ± 0.98 | **1.39** ± 0.92 | **3.17** ± 1.10 | 9.09 ± 2.10 | **10.46** ± 1.73 |

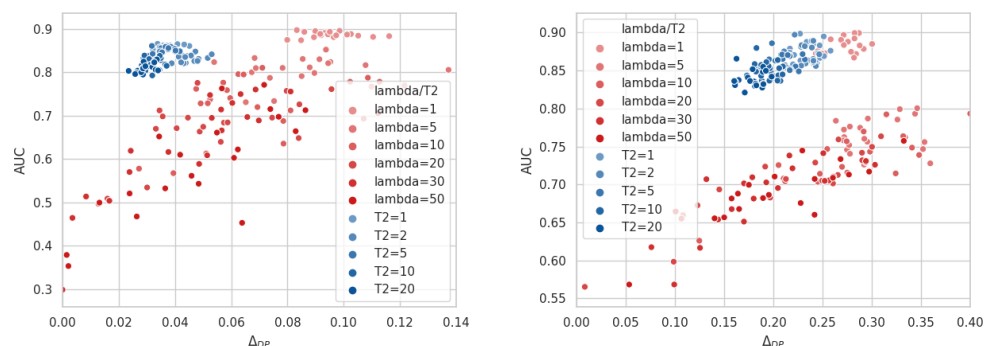

Figure 6: Compare to adversarial training on vertex representations. Left: *Oklahoma97*; Right: *Cora*.

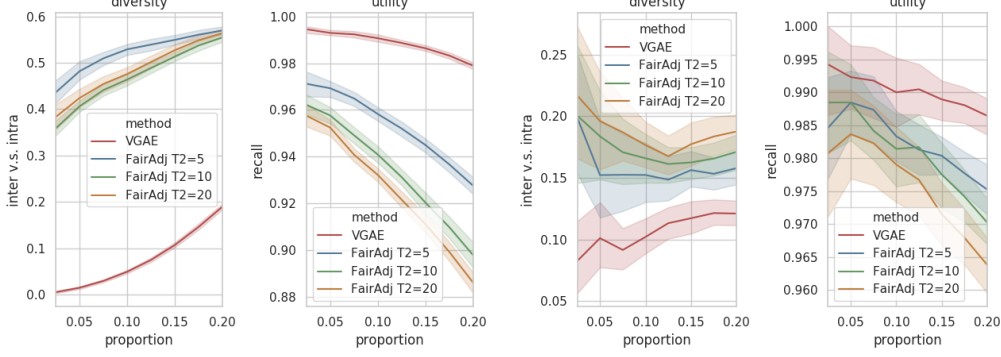

Figure 7: Diversity and utility in recommendations. Left: *Oklahoma97*; Right: *Cora*.

Table 7: Experimental results on *Facebook#1684*.

| Method | AUC | AP | $\Delta_{\text{DP}}$ | $\Delta_{\text{true}}$ | $\Delta_{\text{false}}$ | $\Delta_{\text{FNR}}$ | $\Delta_{\text{TNR}}$ |
|---|---|---|---|---|---|---|---|
| VGAE | $94.66 \pm .55$ | $93.91 \pm .68$ | $2.03 \pm .81$ | $0.59 \pm .49$ | $\mathbf{0.90} \pm .57$ | $4.48 \pm 1.57$ | $4.94 \pm 1.32$ |
| node2vec | $90.57 \pm .74$ | $85.61 \pm 1.09$ | $\mathbf{1.70} \pm 1.43$ | $\mathbf{0.52} \pm .49$ | $2.47 \pm 1.52$ | $6.51 \pm 2.04$ | $5.06 \pm 1.36$ |
| fairwalk | $90.56 \pm .63$ | $85.58 \pm .87$ | $1.97 \pm 1.51$ | $0.62 \pm .47$ | $2.14 \pm 1.77$ | $6.92 \pm 2.19$ | $5.03 \pm 1.46$ |
| FairAdj$_{\text{T2}=1}$ | $\mathbf{94.68} \pm .48$ | $\mathbf{93.94} \pm .62$ | $2.02 \pm .82$ | $0.60 \pm .50$ | $0.93 \pm .60$ | $\mathbf{4.42} \pm 1.57$ | $\mathbf{4.82} \pm 1.54$ |
| FairAdj$_{\text{T2}=20}$ | $94.63 \pm .49$ | $93.84 \pm .64$ | $1.77 \pm .81$ | $0.53 \pm .41$ | $0.92 \pm .49$ | $5.00 \pm 1.52$ | $4.86 \pm 1.41$ |

## E   FAIR VERTEX REPRESENTATION

As an intermediate result, we inspect the fairness in vertex representation in Figure 8. To quantify that, we conduct K-means clustering on vertex representation and evaluate the ratio of samples from different sensitive groups within each clusters, and the ratio is called balance. We range the number of clusters from 4 to 8 and report the average balance across all clusters. In general, the higher the balance, the fairer in vertex representations. Overall the series of **FairAdj** achieves a higher balance, which shows the invariant representations on vertices across different sensitive groups.

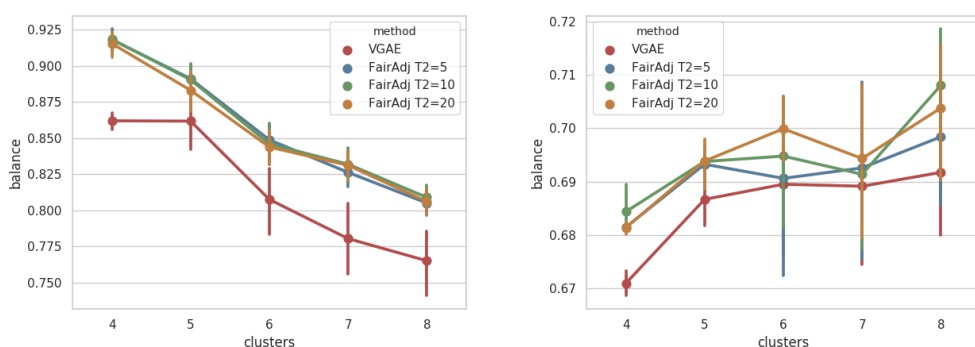

Figure 8: Evaluations on balance of clusters. Left: *Oklahoma97*; Right: *UNC28*.

## F   EXTEND COROLLARY 4.1 TO TWO-LAYER GNNS

For $\Delta_{DP}^{(2)}$ on vertices after passing two layer $\text{GNN}_{\theta}^{(2)}(X, \widetilde{A}) := (\widetilde{A}(\text{GNN}_{\theta}^{(1)}(X, \widetilde{A}))W_{\theta}^{(2)}) = \rho(\widetilde{A}\rho(\widetilde{A}XW_{\theta}^{(1)})W_{\theta}^{(2)})$. Here we denote $\mu_i' := \mathbb{E}_{v \sim U}[\text{GNN}_{\theta}^{(1)}(v, \widetilde{A}) \mid v \in S_i]$, $i = 0, 1$, $Q' := \sup\{\|\text{GNN}_{\theta}^{(1)}(v, \widetilde{A})\|_2 \mid v \in \mathcal{V}\}$. $\mu_i'' := \mathbb{E}_{v \sim U}[\text{GNN}_{\theta}^{(2)}(v, \widetilde{A}) \mid v \in S_i]$, $i = 0, 1$, $Q'' := \sup\{\|\text{GNN}_{\theta}^{(2)}(v, \widetilde{A})\|_2 \mid v \in \mathcal{V}\}$. Let $\sigma_1$ be the maximal deviation of $\{v | v \in \mathcal{V}\}$, and $\sigma_2$ be the maximal deviation of $\{\text{GNN}_{\theta}^{(1)}(v, \widetilde{A}) | v \in \mathcal{V}\}$. We have $Q' \leq L\|W_{\theta}^{(1)}\|_2 Q$, $Q'' \leq L\|W_{\theta}^{(2)}\|_2 Q' \leq L^2\|W_{\theta}^{(1)}\|_2\|W_{\theta}^{(2)}\|_2 Q$.

Then

$$\Delta_{DP}^{(2)} \leq Q''\|\Sigma\|_2 \cdot \|\mu_0'' - \mu_1''\|_2 \leq QL^3\|W_{\theta}^{(2)}\|_2^2\|W_{\theta}^{(1)}\|_2(\alpha\|\mu_0' - \mu_1'\|_2 + 2\sqrt{M}\sigma_2)$$

And

$$\|\mu_0' - \mu_1'\|_2 \leq L\|W_{\theta}^{(1)}\|_2(\alpha\|\mu_0 - \mu_1\|_2 + 2\sqrt{M}\sigma_1)$$

Finally we have:

$$\Delta_{DP}^{(2)} \leq QL^4\|W_{\theta}^{(2)}\|_2^2\|W_{\theta}^{(1)}\|_2^2\alpha^2\|\mu_0 - \mu_1\|_2 + 2\sqrt{M}QL^3\|W_{\theta}^{(2)}\|_2^2\|W_{\theta}^{(1)}\|_2(L\|W_{\theta}^{(1)}\|_2 \cdot \sigma_1 + \sigma_2)$$

