# OpenReview forum: "On Dyadic Fairness: Exploring and Mitigating Bias in Graph Connections"
_ICLR.cc/2021/Conference — ICLR 2021 Poster_

### Official Review · AnonReviewer2 · 2020-10-27

**Rating:** 5
**Confidence:** 3

**Review:**

This paper considers the problem of performing link prediction in a way that satisfies demographic parity through Graph Neural Networks. The authors propose an algorithm to modify the weights on the graph adjancency matrix such that the resulting GNN-based link predictions are satisfy the fairness definition. They evaluate this algorithm on several real-world datasets. The experiments demonstrate that the proposed algorithm increases fairness as measured by demographic parity at the expense of accuracy.

This paper is quite dense and difficult to read. There is little exposition to provide the intuition for what the authors are doing, especially for the reader who is unfamiliar with the particular techniques used. The motivation is also not completely clear to me -- while there are definitely instances where people are interested in demographic parity as a constraint, I don't necessarily see how link prediction in particular is one of those instances.

From a technical perspective, the work appears to be novel and sound. The experiments show that the proposed algorithm achieves what it sets out to do, in comparison with baselines.

---

> ### Author Response · Authors · 2020-11-18
> **Response to Reviewer 2**
>
> Thanks for your comments. We have polished the description for dyadic fairness in Section 3 to present a straightforward understanding for readers.
>
> ---
> The motivation of this paper comes from the potential bias in user recommendation in social networks and recommendation systems. In social networks, users with the same gender or religion are more likely to connect with others. A data-driven model may capture the correlations between user connections and their sensitive attribute, and continue to recommend users with the same sensitive attributes, thus leading to segregation in gender or religion. We propose our model to alleviate this kind of bias.
>
> First, we formulate what is the fairness in link prediction in a homogeneous graph with the help from demographic parity [1,2,3,4,5]. Demographic parity, a well-used fairness notion in classification problems, says given two sensitive groups (e.g., male/female), it encourages the classifier to give positive outcomes to the two groups at the same rate. In our problem, we treat the intra link and inter link as two sensitive groups (male-male and female-female/male-female), and also ask to equalize the outcomes of these two types of links. Having such a fairness notion, ideally, from a collection of potential links, we are able to see intra links and inter links at the same rate, which break segregation by encouraging more connections across sensitive groups (more connections between males and females). This is the problem formulation and why we need to have dyadic fairness to unbias the link predictions.
>
> [1] Johndrow, James E., and Kristian Lum. "An algorithm for removing sensitive information: application to race-independent recidivism prediction." The Annals of Applied Statistics 13.1 (2019): 189-220.
> [2] Kamiran, Faisal, and Toon Calders. "Classifying without discriminating." 2009 2nd International Conference on Computer, Control and Communication. IEEE, 2009.
> [3] Louizos, Christos, et al. "The Variational Fair Autoencoder." ICLR. 2016.
> [4] Madras, David, et al. "Learning adversarially fair and transferable representations." arXiv preprint arXiv:1802.06309 (2018).
> [5] Zemel, Rich, et al. "Learning fair representations." International Conference on Machine Learning. 2013.

---

### Official Review · AnonReviewer3 · 2020-10-28
**Clarifications needed to understand the objective and assumptions**

**Rating:** 5
**Confidence:** 3

**Review:**

Quality:
The work overall is of "okay" quality, my concerns are listed at the end of this review.

Clarity:
I believe the presentation can be improved. Specifically, the problem setting can be better explained in the introduction, where terms like "dyadic fairness" or "homogeneous graph" are thrown without much details. Perhaps a practical example can help understanding the problem under analysis.

Significance:
Link prediction is an important problem in the analysis of social/citation networks, thus, the topic of the work is relevant to the community. In addition, the problem is analyzed under fairness considerations which increases the significance.
The results, however, are not strong in my opinion (at least from my first review of the paper).

Questions:
My next questions might come from misunderstandings, in which case I would like to clarify:

* Why is homogeneous graph said to be the focus of the analysis in the introduction but, later on, it is not even mentioned in the technical results. Was that assumption being used? did it matter at all?
* Authors state that the goal is to have link predictions that are independent of the sensitive attributes. Why does minimizing the norm of the difference of averages of node vectors of each group imply predictions being independent of the sensitive attribute?
Not having a good explanation for this would undermine the work a lot.
* The references in Section 2 seem rather arbitrary, for instance, in "fair machine learning" I do not see the work "learning fair representations" (Zemel et al. 2013). I wonder why it was not referred or rather why are the listed references there preferred over other works?

---

> ### Author Response · Authors · 2020-11-18
> **Response to Reviewer 3**
>
> Thanks for reviewing our paper.
>
> ---
> **1. Homogeneous graph**
> A  homogeneous graph contains vertices with the same type. For instance, vertices in a homogeneous social network present users.  Since fairness in link prediction is a newly emerged topic, for the analysis and the method we focus on homogeneous graphs in this paper for simplicity. All datasets we use for experiments are homogeneous graphs. It would be interesting to extend our work to the heterogeneous graph. However,  there might be additional work on problem formulation on the extension to heterogeneous graph, since different types of vertices have different types of sensitive attributes. For example, in heterogeneous social networks, there are two types of vertices, user and tweet. Gender can be selected as the sensitive attributes for users, while political opinion propensity can present the sensitive attributes for tweets. We can have the fairness if we want different genders to receive equal opinions from political opinions.
>
> ---
> **2. From dyadic fairness to Proposition 4.1**
> We have refined Section 3 to present a more clear formulation of dyadic fairness. We get the notion demographic parity [1,2,3,4,5], a well-used fairness notion in classification, to help us formulate dyadic fairness in the link prediction problem. In classification problems, demographic parity says given two sensitive groups (e.g., male/female), we want the classifier to give positive outcomes to the two groups at the same rate (e.g. no gender discrimination in loan applications). In our problem, we treat intra link (male-male/female-female) and inter link (male-female) as two sensitive groups in the content of demographic parity, and also ask to equalize the outcome of these two types of links. Having such a fairness notion, ideally, from a collection of potential links, we can get intra links and inter links at the same rate, which can break segregation by encouraging more connections across sensitive groups (more connections between males and females). By measuring the absolute difference between the predictive scores on intra and inter links (right side in proposition 4.1), and based on proposition 4.1, we are safe to reduce the achievement of dyadic fairness to the minimization of the norm gap between groups.
>
> [1] Johndrow, James E., and Kristian Lum. "An algorithm for removing sensitive information: application to race-independent recidivism prediction." The Annals of Applied Statistics 13.1 (2019): 189-220.
> [2] Kamiran, Faisal, and Toon Calders. "Classifying without discriminating." 2009 2nd International Conference on Computer, Control and Communication. IEEE, 2009.
> [3] Louizos, Christos, et al. "The Variational Fair Autoencoder." ICLR. 2016.
> [4] Madras, David, et al. "Learning adversarially fair and transferable representations." arXiv preprint arXiv:1802.06309 (2018).
> [5] Zemel, Rich, et al. "Learning fair representations." International Conference on Machine Learning. 2013.
>
> ---
> **3. Extension on related works**
> Thanks for providing this paper. In our previous version,  we mainly introduced some recent works that use neural networks to find a fair representation, which is closer to our approach (graph neural networks). According to your suggestion, we have extended the discussion in fair machine learning in Section 2 and added the following references in our updated version.
>
> [1] Zemel, Rich, et al. "Learning fair representations." International Conference on Machine Learning. 2013.
> [2] Kleinberg, Jon, Sendhil Mullainathan, and Manish Raghavan. "Inherent Trade-Offs in the Fair Determination of Risk Scores." arXiv preprint arXiv:1609.05807 (2016).
> [3] Beutel, Alex, et al. "Data Decisions and Theoretical Implications when Adversarially Learning Fair Representations." (2017).
> [4] Louizos, Christos, et al. "The Variational Fair Autoencoder." ICLR. 2016.
> [5] Zafar, Muhammad Bilal, et al. "Fairness constraints: Mechanisms for fair classification." Artificial Intelligence and Statistics. PMLR, 2017.
> [6] Edwards, Harrison, and Amos Storkey. "Censoring representations with an adversary." arXiv preprint arXiv:1511.05897 (2015).

---

### Official Review · AnonReviewer1 · 2020-10-28
**Interesting paper for understanding graph representation fairness and an alternative strategy to mitigate the potential bias**

**Rating:** 7
**Confidence:** 3

**Review:**

##########################################################################

Summary:

This work studied how the connections in graph data affect 'dyadic fairness' when applying GNN for representation learning on homogenous graphs. Inspired by the theoretical findings, the authors proposed a method (FairAdj) on top of VGAE to adjust adjacency weight matrix as a separate step to address the fairness-utility trade-off. The effectiveness of the proposed method is demonstrated through extensive experiments on different real-world datasets.

##########################################################################

Strength:

- Fairness of graph representation learning is a relevant and emerging research topic
- The analysis and the proposed method are technically sound and the experiments are executed well
- Many interesting insights are obtained from the theoretical analysis

##########################################################################

Weakness:

- The overall presentation can be further improved.
- Current analysis and method are conducted at homogeneous graph only
- Efficiency / Time complexity analysis is not provided

##########################################################################

Detailed Comments and Questions:

This is an overall good work which contributes some interesting insights around graph representation fairness, a solid method to mitigate potential bias, and a series quantitative experiments to demonstrate its effectiveness. I also enjoyed reading this work and felt inspired. Overall I vote for *accept* but there are also a few points which could be further improved.

- The overall presentation is a bit dry and tough to digest. Specifically it's a bit difficult to understand dyadic fairness from high-level and differentiate this concept with other algorithmic fairness work on graph data. Might be helpful to highlight the formal definition of this and the corresponding insights.

- Current analysis and method are conducted at homogeneous graph and the analysis is concluded on 1-layer GNN scenario, would be better to open discussions for future work

- One missing part of the proposed mitigation strategy is around the time efficiency analysis. For practical reason, how its efficiency (specifically the two-step iteration for adjusting the adjacency matrix) compared to a typical adversarial approach?

- What I'm further curious about is how the theoretical findings can be extended to 1) multi-hop/layer GNN and 2) bipartite graph such as the typical user-item recommendation setting. For 2) seems not very obvious; for 1) I did see a great practical value as in real-world applications it is very common to go beyond 1-layer network (which is also a way to obtain superior utility), specifically what does corollary 4.1 imply? does that mean with small L, we are expecting to see tighter upper bound with the network goes deeper?

##########################################################################

Typos:

- Remark 1: "linke" -> "link"

---

> ### Author Response · Authors · 2020-11-18
> **Response to Reviewer 1**
>
> Happy to know you are enjoying our paper!
>
> ---
> **1. Formulation of dyadic fairness**
> We update a formal definition with more interpretation of dyadic fairness in Section 3 in our paper. We establish dyadic fairness from demographic parity, a well-used fairness notion in a series of literature [1,2,3,4,5]. In the classification problem, demographic parity says given two sensitive groups (e.g., male/female), we want the classifier to give positive outcomes to the two groups at the same rate (e.g. bank loan access with no gender discrimination). In our problem, we treat the intra links (e.g., male-male/female-female) and inter links (e.g., male-female) as two sensitive groups in demographic parity formulation, and also ask to equalize the predictive scores over these two groups, therefore run into the definition 3.1 of dyadic fairness in our updated paper. Ideally, from a bag of potential links, we can get intra and inter links at the same rate by achieving dyadic fairness, hence break segregation by encouraging more connections across sensitive groups (e.g., groups of male and female). In [6] they formulate demographic parity at all types of links, while we only consider intra and inter links in our work.
>
> [1] Johndrow, James E., and Kristian Lum. "An algorithm for removing sensitive information: application to race-independent recidivism prediction." The Annals of Applied Statistics 13.1 (2019): 189-220.
> [2] Kamiran, Faisal, and Toon Calders. "Classifying without discriminating." 2009 2nd International Conference on Computer, Control and Communication. IEEE, 2009.
> [3] Louizos, Christos, et al. "The Variational Fair Autoencoder." ICLR. 2016.
> [4] Madras, David, et al. "Learning adversarially fair and transferable representations." arXiv preprint arXiv:1802.06309 (2018).
> [5] Zemel, Rich, et al. "Learning fair representations." International Conference on Machine Learning. 2013.
> [6] Rahman, Tahleen A., et al. "Fairwalk: Towards Fair Graph Embedding." IJCAI. 2019.
>
> ---
> **2. Beyond homogeneous graph and extend the analysis from 1-layer GNNs**
> Since fairness in link prediction is a newly emerged topic, we focus on homogeneous graphs in this paper for simplicity. It would be interesting to extend our work to the heterogeneous graphs. However, there might be additional work on problem formulation on the extension, since different types of vertices will come to have different types of sensitive attributes. A potential case in heterogeneous social networks is, there are two types of vertices, user and tweet. Gender can be selected as the sensitive attributes for users, while political opinion propensity can present the sensitive attributes for tweets. We may want different genders are receiving equal political opinions.
>
> We provide an extended theoretical form of two-layer GNNs in our updated appendix since two-layer in GNNs are most widely used.
>
> ---
> **3. Time efficiency analysis**
> In our method, the step for adjusting the adjacency matrix is O(NDlogD) where N is the number of vertices and D is the average degree of vertices in the graph. The rest of our method (step for optimizing GNNs) is linear to the size of model parameters and is the same in a typical adversarial approach.
>
> ___
> **4. Extension for theoretical findings**
> We provide a theoretical form of two-layer GNNs in our updated appendix, for GNNs with two-layer are widely used and have the most practical value. A two-layer GNNs can be viewed as two-hop since it receives the message from second-hop neighbors.
>
> Our analysis is feasible for any homogeneous bipartite graph since we do not put any assumptions on the graph connections in the theoretical part. If in a user-item setting, we need to handle the types of sensitive attributes as discussed in **2**.
>
> Corollary 4.1 implies the algebraical relationship between the upper bound of transformed dyadic fairness after passing a real GNN layer and the model parameters. Both Corollaries on 1-layer GNN and 2-layer GNN have upper bounds composed of two parts: one is the linear mapping of original dyadic fairness, $|\mu_0-\mu_1|$, and another one is additional errors $\sigma$ from maximal deviations within groups.  If the error term is relatively ignorable to  $|\mu_0-\mu_1|$, a contractive upper bound could approximately be achieved with L and spectral norm on $W_\theta$ less than one.
>
> ---
> Thanks for the careful check on typos, revised in the updated paper.

---

### Official Review · AnonReviewer4 · 2020-10-30
**Review of dyadic fairness**

**Rating:** 7
**Confidence:** 3

**Review:**

This paper discusses fairness problems in graph embedding for link prediction to mitigate problems such as graph segregation related to sensitive attributes. The paper uses a variational graph autoencoder to learn an embedding, and predicts links based on distance in this embedded space. The notion of dyadic fairness refers to statistical constraints on link prediction between/across sensitive attribute groups, such as parity for positive predictions or false negatives. The learning algorithm updates a weighted adjacency matrix with nonzero entries preserving the observed adjacency structure of the graph, while optimizing edge weights to tradeoff between utility and dyadic fairness of edges predicted from the weights. The paper compares the proposed method to a baseline and a competing fair graph embedding method on several example datasets to show good performance on utility, fairness, and diversity of inter/intra-group link predictions.

I recommend accepting this paper because it focuses on a problem where there is relatively little previous work, introduces a novel approach, and shows empirically that the approach is competitive. I have a few questions or comments which, if addressed, could improve my rating of the paper.

0. Is it really necessary to constrain the nonzero entries of the adjacency matrix based on observed edges? The paper suggests "adding fictitious links might mislead the directions of message passing," but is there any reason to believe the resulting errors would be any worse than errors from other sources of uncertainty in the problem? It seems possible to me, a priori, that an algorithm without this constraint might achieve both better utility and fairness under some conditions on some examples. This might be worth exploring or mentioning, even if the present paper makes the design choice of keeping this constraint.

1. It is not clear to me exactly how the theoretical findings motivate the algorithm. Perhaps this would be clearer with a better explanation related to my previous point about the structural constraint.

2. In the theoretical analysis, immediately after Corollary 4.1, the paper states "Multiple layers of GNNs can be reasoned out similarly." This seems like quite a stretch without providing further justification.

3. It is a little confusing to determine the meaning of "proportion" on the horizontal axis in Figure 2.

4. For the experiments, it's not clear to me why the category of an article should be considered a sensitive attribute in citation networks. Perhaps there is an explanation of the fairness concern in citation networks that would make this more obvious. Instead of so many citation networks, the experiments might be improved with social networks having sensitive attributes like race, religion, or political identification, where there are social science reasons to be especially concerned about social network segregation. I believe there is less reason to believe there will be a high degree of network segregation on the basis of gender than on the basis of those other sensitive attributes, so I would suggest not limiting the sensitive attribute to only gender in the social network examples.

---

> ### Author Response · Authors · 2020-11-18
> **Response to Reviewer 4**
>
> We really appreciate your insightful feedback.
>
> ---
> **0.  The structural constraint**
> We agree that removing the constraint and adding more variables for optimization may further optimize the objective function with an extended feasible region. However, message passing in graph neural networks follows the graph connections, and it helps smooth the discriminability among linked users and find the pattern behind graph connections. If we manually build some fictitious links, an instance may start to receive information from other instances that previously it does not connect to. Consequently, the representation of this instance may be polluted and further hurt the predictive utility. We have an experiment on *Cora* without this structural constraint, and we find with similar fairness it gives worse result in utility
>
> |Method |AUC $\uparrow$ |AP $\uparrow$ |$\Delta_{\text{DP}} \downarrow$ |$\Delta_{\text{true}} \downarrow$|$\Delta_{\text{false}} \downarrow$|$\Delta_{\text{FNR}} \downarrow$|$\Delta_{\text{TNR}} \downarrow$|
> |:---|:---:|:---:|:---:|:---:|:---:|:---:|:---:|
> |FairAdj_T2=20 w/ constraint | $\small{83.85}\scriptsize{\pm 1.07}$ | $\small{86.08}\scriptsize{\pm 0.93}$ | $\small{17.87}\scriptsize{\pm 1.18}$ | $\small{5.40}\scriptsize{\pm 2.23}$ | $\small{3.74}\scriptsize{\pm 1.46}$ | $\small{16.75}\scriptsize{\pm 4.87}$ | $\small{15.37}\scriptsize{\pm \text{3.84}}$ |
> | FairAdj_T2=20 w/o constraint | $\small{81.33}\scriptsize{\pm 1.32}$ | $\small{84.82}\scriptsize{\pm 1.59}$ | $\small{17.97}\scriptsize{\pm 1.33}$ | $\small{7.85}\scriptsize{\pm 1.72}$ | $\small{2.64}\scriptsize{\pm 1.10}$ | $\small{20.87}\scriptsize{\pm 4.82}$ | $\small{13.24}\scriptsize{\pm \text{3.92}}$ |
>
> Another point for preventing fictitious links is the scalable issue. Adding those variables will break the sparsity of the graph and turn it into a complete graph in optimization, making the total size of variables quadratic towards the number of nodes, and disable the fast sparse matrix multiplication.
>
> ---
> **1.  Theoretical findings in the algorithmic design**
> Our theoretical findings show regulating weights on graph connections can help to achieve dyadic fairness in predictive outcomes, but this solution cannot guarantee arbitrary fairness due to the resistance if preserve original graph connections and may suffer an additional error from deviation. Remove the structural constraint can help us remove the lower bound in Theorem 4.1 and achieve arbitrary fairness, but will run into the scenario as described in the above answer. In the FairAdj algorithm, we design to regulate the weights on graph connections by empirically minimizing the difference in predictive scores between intra and inter links.
>
> ---
> **2. Further justification on Corollary 4.1**
> We provide a form of two-layer GNNs in the updated appendix, for GNNs with two-layer are most widely used. The result is given by using the output from one-layer GNNs and go through Corollary 4.1 again.
>
> ---
> **3. 'Proportion' in Figure 2**
> Proportion 0.05 in Figure 2 means that we check the link prediction with scores in the top 5% and investigate how diverse they are as shown on the vertical axis. Other values in proportion follow the same manner. We have added more description in Figure 2 caption for clarity.
>
> ---
> **4. Citation networks in experiments**
> Considering some research topics in the recent era, that actually there are no clear boundaries between, for example, machine learning, data mining, pattern recognition, and other related domains.  Articles in _Cora_ dataset used in our experiments are labeled with “reinforcement learning”, “neural networks”, “probabilistic methods”, and these domains are overlapping. We always want to be inspired by other works but not be constrained into a single area. In our views, these domains could be treated equally in scholarly article recommendations, just like genders in social networks. Breaking the research boundary by recommending related articles from other fields can promote the interdisciplinary community.
> We agree that gender may not be a severe issue in segregation compare to other sensitive attributes. If there is a suitable dataset with political identification, our method can be easily applied as well.

---

### Author Response · Authors · 2020-11-18
**Response updated**

We would like to thank all the feedback that helps us improve our paper. Please find the point-to-point response below.

---

### Author Response · Authors · 2020-11-25
**Paper updated**

We update our paper as follows according to the feedback
1. We extend the discussions and add more references in related work 'fair representation learning' part to provide readers a more comprehensive background.
2. We update Section 3 Preliminaries and add a more concrete description as well as a mathematical formulation on the idea of dyadic fairness. Also, we discuss how our definition is related to demographic parity but different from it due to the problem content.
3.  We update the caption in Figure 2 to better illustrate the meaning of the x-axis. Results in this figure also demonstrate that our model can bring more diverse recommendations in the outcome.
4. In the newly added section Appendix E, we extend the form Corollary 4.1 into a two-layer GNNs.

---

### Decision · Program_Chairs · 2021-01-07
**Final Decision**

**Decision:**

Accept (Poster)

**Comment:**

This paper is devoted to "dyadic fairness" in representation learning. All the reviewers agreed that the contribution is novel, original and technically sound. However, all the reviewers agreed that the paper should be improved in terms of presentation -- for two reviewers, presentation/clarity issues were at the core of their weak rejects. The most positive reviewers highlighted that the problem is still understudied despite the flurry of work on fair machine learning in the last years and therefore the contribution deserves to be accepted. If there is room, this paper can be accepted as a poster.